# Greater fuel efficiency is potentially preferable to reducing NO$_x$ emissions for aviation's climate impacts

Agnieszka Skowron [1✉], David S. Lee [1], Rubén Rodríguez De León [1], Ling L. Lim [1] & Bethan Owen [1]

Aviation emissions of nitrogen oxides (NO$_x$) alter the composition of the atmosphere, perturbing the greenhouse gases ozone and methane, resulting in positive and negative radiative forcing effects, respectively. In 1981, the International Civil Aviation Organization adopted a first certification standard for the regulation of aircraft engine NO$_x$ emissions with subsequent increases in stringency in 1992, 1998, 2004 and 2010 to offset the growth of the environmental impact of air transport, the main motivation being to improve local air quality with the assumed co-benefit of reducing NO$_x$ emissions at altitude and therefore their climate impacts. Increased stringency is an ongoing topic of discussion and more stringent standards are usually associated with their beneficial environmental impact. Here we show that this is not necessarily the right direction with respect to reducing the climate impacts of aviation (as opposed to local air quality impacts) because of the tradeoff effects between reducing NO$_x$ emissions and increased fuel usage, along with a revised understanding of the radiative forcing effects of methane. Moreover, the predicted lower surface air pollution levels in the future will be beneficial for reducing the climate impact of aviation NO$_x$ emissions. Thus, further efforts leading to greater fuel efficiency, and therefore lower CO$_2$ emissions, may be preferable to reducing NO$_x$ emissions in terms of aviation's climate impacts.

[1] Faculty of Science and Engineering, Manchester Metropolitan University, Manchester, UK. ✉email: a.skowron@mmu.ac.uk

Emission standards for aircraft $NO_x$ are set by the Committee on Aviation Environmental Protection of the International Civil Aviation Organisation (ICAO-CAEP). In the past, aircraft $NO_x$ emissions standards have been set to protect local air quality and have been assumed to have co-benefits for climate protection, as aircraft $NO_x$ results in an overall warming effect at present[1,2]. Emissions of $NO_x$, whether from aviation or other sources, result in the short-term formation of ozone ($O_3$) (a warming) and the long-term destruction, via hydroxyl (OH) production, of small amounts (~a few percent) of ambient methane ($CH_4$) (a cooling)[1]. In addition, the methane reduction results in a long-term reduction in $O_3$ (cooling)[3] and a long-term reduction in $H_2O$ in the stratosphere (cooling) from reduced oxidation of methane[4]. The net balance of these components ranges from positive for aviation $NO_x$, to negative for shipping and surface $NO_x$[4–6].

The general scientific advice given to the ICAO-CAEP to date has been to reduce emissions of both $NO_x$ and $CO_2$. However, reducing both is problematic because of a technological trade-off between aviation $NO_x$ and $CO_2$[7–9]. Furthermore, one has to be very careful trading short-lived climate forcers against long-lived greenhouse gases, e.g. the reduction of $NO_x$ emissions might result in a fuel penalty that in fact can lead to a net climate disbenefit[10]. Here, we present a new analysis of future aviation emission scenarios and the balance of $NO_x$ from surface and aircraft sources, that makes this recommendation uncertain in terms of climate benefits and we suggest that the scientific evidence for reducing aircraft $NO_x$ needs to be revisited.

At subsonic aircraft cruise altitudes of 8–12 km, the atmosphere is sensitive to aircraft $NO_x$ emissions where $O_3$ production is four times more efficient than near the ground[11] and the aviation net $NO_x$ effect also depends on the state of the atmosphere into which $NO_x$ is emitted[12]. Changes in emissions of any surface source that take place as a result of various air quality and climate policies may have impacts on background conditions, and consequently might have an impact on the climate effect of aviation $NO_x$ emissions, which is not independent of background conditions. The changes in the tropospheric composition and global radiative forcing (RF) for various scenarios of anthropogenic $O_3$ precursor emissions have been widely explored[13–15]. However, the impact of changing surface emissions on aircraft $NO_x$ climate effects has been virtually left out of discussions and sensitivity experiments can only be found in one study[3]. This present study aims to fill this gap and start the discussion on how future anthropogenic background emissions can affect the aviation climate impact.

Using a suitable three-dimensional chemistry transport model (CTM) of the global atmosphere (MOZART3)[16,17], we examine the changes in the tropospheric composition and the net RF from aviation $NO_x$ emissions for 30% reductions in the most recent present-day inventories available (2006) of $O_3$ precursor emissions ($NO_x$, carbon monoxide - CO, non-methane volatile organic compounds - NMVOC) and for a future (2050) range of Representative Concentration Pathways (RCP) scenarios together with ICAO-CAEP aviation emission projections (see Methods for details of models used and simulations). These simulations allow an analysis of the relative benefits to reducing aviation impacts from reducing aviation and background $NO_x$ emissions.

## Results

**Aviation net $NO_x$ radiative forcing in 2050.** The resulting RFs (Table 1) highlight that an aviation net $NO_x$ RF can vary greatly depending on the background condition, and both anthropogenic surface and aircraft emissions affect the aviation net $NO_x$ RF. At present, all scenarios predict increased aircraft $NO_x$ emissions in the year 2050 that reach 2.17 Tg(N) year$^{-1}$ for the low air-traffic

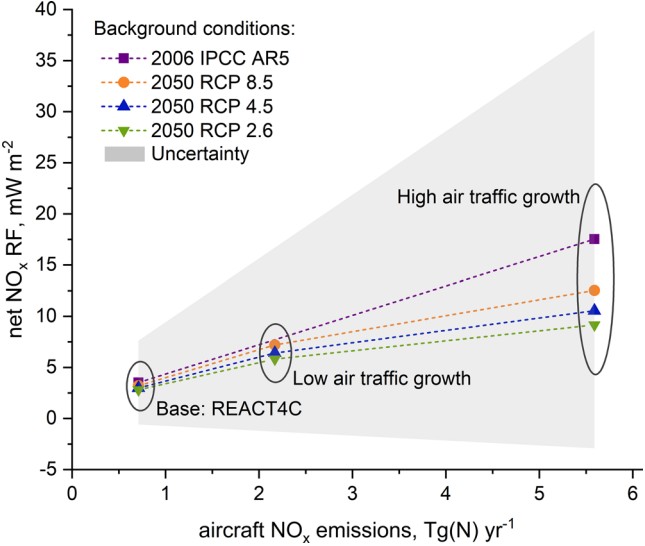

**Fig. 1 Net $NO_x$ radiative forcing by emission rate, original $CH_4$ parameterisation.** Aviation net $NO_x$ radiative forcing (RF) (the sum of the short-term positive $O_3$ RF perturbation and the negative RF terms caused by a reduction in $CH_4$ lifetime, see Methods), by aviation $NO_x$ emission rate according to a range of background emission scenarios, utilising the IPCC Fifth Assessment Report[59] simplified expression for the calculation of $CH_4$ forcing. Aviation net $NO_x$ RF systematically increases with increasing $NO_x$ emissions from aviation, showing a variation according to the background surface emissions, with high mitigation (RCP 2.6) having a smaller aviation net $NO_x$ RF than, lower mitigation scenarios (RCP 4.5, RCP 8.5) for the same aviation $NO_x$ emission. Overall uncertainties are indicated by the grey shading, which is one standard deviation (68% confidence interval) from the ensemble of 20 $NO_x$ studies presented in Supplementary Fig. 1.

growth and optimistic technology development and 5.59 Tg(N) year$^{-1}$ for high air-traffic growth and low technology development: which compared with the year 2006, 0.71 Tg(N) year$^{-1}$, are significant increases. In contrast, reductions of surface $O_3$ precursor emissions are projected under each of the applied RCP scenarios for the year 2050, for which the cleanest background is predicted under RCP 2.6. The total aircraft net $NO_x$ RF in 2006 is 3.5 mW m$^{-2}$ and it increases to between 5.8 and 12.5 mW m$^{-2}$ in 2050 for low- and high-growth scenarios, respectively; within each RCP scenario, the net aviation $NO_x$ RF can differ by ~23%, depending on background conditions. The largest aviation net $NO_x$ RF is observed under RCP 8.5 and the smallest for RCP 2.6 (for equal aviation $NO_x$ emissions). This significant increase in the aviation net $NO_x$ RF for 2050 is driven mainly by an intense rise of aviation $NO_x$ emissions; however, reduced surface emissions resulting in a cleaner background atmosphere in the year 2050 to some extent mitigates the aviation impact (Fig. 1). For instance, if the anthropogenic surface emissions were kept constant at present levels, the 2050 aircraft net $NO_x$ RF of the high air-traffic growth would be 17.5 mW m$^{-2}$, that is 48% greater than under the 2050 RCP 2.6 and 28% greater than under 2050 RCP 8.5 background conditions.

This observed increase of aviation net $NO_x$ RFs in the future is in agreement with other studies that have explored the climate impact from aviation $NO_x$ emissions in 2050. Global CTMs and chemistry-climate models (CCMs) using 2050 aviation emissions derived from the Aviation Environmental Design Tool (AEDT) and the RCP 4.5 background scenario have been employed[18–20]. Unger et al.[19] calculated that both the positive short-term $O_3$ and negative $CH_4$ RFs in 2050 increased by ~80% for the AEDT Base scenario (4.0 Tg(N) year$^{-1}$), whereas Khodayari et al.[20] estimated

**Table 1 Aircraft $NO_x$ radiative forcing (RF, mW m$^{-2}$) for different background and aircraft emission scenarios in 2006 and 2050. Net $NO_x$ RF is a sum of a short-term $O_3$ ($sO_3$), $CH_4$, $CH_4$-induced $O_3$ ($IO_3$) and SWV.**

| Emissions | | | RF, mW m$^{-2}$ | | | | |
|---|---|---|---|---|---|---|---|
| | Background | Aircraft | $sO_3$ | $CH_4$ | $IO_3$ | SWV | Net $NO_x$ |
| 2006 IPCC AR5 | Base | REACT4C | 14.8 | −6.9 | −3.4 | −1.0 | 3.5 |
| | −30% $NO_x$ | | 17.7 | −9.5 | −4.8 | −1.4 | 2.0 |
| | −30% CO | | 14.1 | −6.7 | −3.4 | −1.0 | 3.0 |
| | −30% NMVOC | | 13.9 | −6.7 | −3.4 | −1.0 | 2.8 |
| | −30% ALL | | 16.2 | −8.5 | −4.3 | −1.3 | 2.2 |
| 2050 | RCP 8.5 | Low $NO_x$ High Tech | 45.9 | −23.4 | −11.7 | −3.5 | 7.2 |
| | RCP 4.5 | | 43.7 | −22.6 | −11.3 | −3.4 | 6.4 |
| | RCP 2.6 | | 40.6 | −21.1 | −10.5 | −3.2 | 5.8 |
| | RCP 8.5 | High $NO_x$ Low Tech | 96.3 | −50.8 | −25.4 | −7.6 | 12.5 |
| | RCP 4.5 | | 90.7 | −48.6 | −24.3 | −7.3 | 10.6 |
| | RCP 2.6 | | 83.9 | −45.3 | −22.7 | −6.8 | 9.2 |

the 2050 short-term $O_3$ RF to be 48–75% greater than in 2006, and the 2050 $CH_4$ RF increased by 57–80%. From these studies, the available 2050 short-term $O_3$ RF ranged from 30 to 162 mW m$^{-2}$ and the $CH_4$ RF varied from −36 to −72 mW m$^{-2}$ (all estimates are for AEDT-Base and RCP 4.5 scenarios)[18,21]. The 90.7 and −48.6 mW m$^{-2}$ calculated in this study with the RCP 4.5 background emissions are in line with estimates found in the literature. The existing spread in the calculated aviation $NO_x$-induced effects is the result of both differences in the projections of aircraft emissions and the inter-model differences. The latter difference raises an important level of uncertainty[3], for which the differences in model chemistry schemes and the treatment of physical processes play an important role. Also, due to the inclusion of more feedback processes and coupled interactions (particularly aerosol and cloud coupling processes), different responses between the offline models (CTMs) and the fully coupled models (CCMs) can be observed[18]. However, based on the reported net $NO_x$ RFs available in the literature any systematic differences between CTMs and CCMs cannot be identified (Supplementary Fig. 1 and Supplementary Note 2).

**The impact of surface emissions on aviation net $NO_x$ radiative forcing.** Not only does the overall reduction of background emissions (i.e. different RCP scenarios) change the aircraft impact (same aircraft emissions), but also the mix of these reductions can have an impact. This is demonstrated by reducing individual precursors by −30% ($NO_x$, CO, NMVOC), and all together (ALL), which changes the oxidative capacity of the atmosphere (Fig. 2). Figure 2 illustrates that a reduction of background $NO_x$ emissions (alone) leads to a significant decrease in hydroxyl radical (OH) concentrations, while the reduction of CO and NMVOC emissions (individually) increases OH concentrations as their oxidation process becomes limited, leading to a reduced production of hydroperoxy radicals ($HO_2$). As a result of the above, the reduction in surface $NO_x$ emissions increases $CH_4$ lifetime and decreases the concentration of tropospheric $O_3$, while the reductions of CO and NMVOC cause the decrease of both tropospheric $O_3$ and $CH_4$, an observation that is consistent with other studies[5,22]. These dependencies are observed not only near the ground but also in the upper troposphere–lower stratosphere (UTLS) region, where most of aviation emissions occur; therefore, affecting aviation effects (Supplementary Fig. 2). In general, reducing surface emissions of $NO_x$, CO and NMVOC individually decreases the net aviation $NO_x$ RF by up to 43%, the most effective being a reduction in surface $NO_x$ emissions alone, and the least effective being a reduction in CO, which results in a reduction of aviation net $NO_x$ RF of 14% (Table 1). A

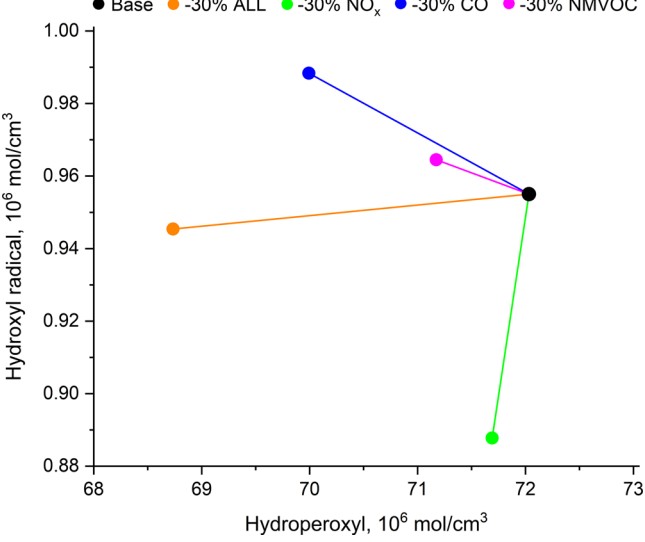

**Fig. 2 The oxidative capacity of the troposphere under different background conditions.** By reducing emissions of individual precursors by 30% ($NO_x$, CO, NMVOC), and all together (ALL) changes the concentrations of hydroxyl radical and hydroperoxyl. The oxidative capacity of the atmosphere to a great extent controls the abundance of most trace gas species, hence affecting $CH_4$ lifetime and concentration of $O_3$ (see text for details). Values are averaged globally within the vertical domain extending from surface to 100 hPa and modelled by MOZART-3 CTM.

simultaneous reduction of all surface emissions by 30% decreases the aviation net $NO_x$ RF by 37% (for the same aircraft emissions). The reduction of surface $NO_x$ has the greatest potential in affecting the aviation net $NO_x$ RF; any 1% change of surface $NO_x$ emissions, modifies aircraft net $NO_x$ RF by ~1.5% (this estimate is inventory-dependant and here it varies from 1.4% to 1.6% for REACT4C 2006 data and high $NO_x$ 2050 scenario, respectively).

There is a well-known increase in $O_3$ production (per unit of emitted N) as background $NO_x$ levels decrease and this is what we observe here as well. However, we also calculate a strong dependence of aircraft $CH_4$ lifetime reduction on surface emissions that becomes more efficient with decreasing $NO_x$. So, for a cleaner $NO_x$ background the positive short-term $O_3$ RF increases, as expected, but the associated $CH_4$ RF (and all the $CH_4$-induced RFs) reduction increases even more, explaining why the net $NO_x$ RF decreases, rather than increasing[3] for a cleaner $NO_x$ background. This strong $CH_4$ response is possibly triggered by increased $CH_4$ lifetime due to reduced oxidative capacity

(Fig. 2). The 30% reduction of surface $NO_x$ increases the positive short-term $O_3$ RF by 16%; however, the magnitude of the negative long-term $CH_4$ RF increases even more, by 28%. Thus, less background $NO_x$ reduces the net $NO_x$ effect from aviation (for the same aviation $NO_x$ emissions). Moreover, it turns out that decreasing surface $NO_x$ emissions plays a larger role in reducing the aviation net $NO_x$ RF than decreasing aircraft $NO_x$ emissions (Fig. 3) in percentage terms. Figure 3 shows a steeper slope in the reduction of net $NO_x$ RF from percentage changes in surface emissions than from aviation emissions themselves. For example, in order to reduce the global climate impact of aviation $NO_x$ by

$1\ mW\ m^{-2}$, a 17% reduction in present levels of surface $NO_x$ emissions is needed; in the case of aviation $NO_x$, it requires the reduction of emissions by 35%. Reducing aviation $NO_x$ emissions by such a large amount (35%) for, e.g. a $1\ W\ m^{-2}$ net $NO_x$ RF reduction could be quite technologically challenging and have a strong risk of increasing aircraft $CO_2$ emissions with a potentially perverse total RF outcome[10]. If a scenario is envisaged of falling surface $NO_x$ emissions, reducing aircraft $NO_x$ emissions at the expense of either missed opportunities to reduce $CO_2$ emissions or even actually increasing $CO_2$ emissions could be exactly the wrong thing to do and induce perverse climate outcomes.

## Discussion

The short-term $O_3$ RF is very sensitive to changes in all explored changes in surface precursor emissions ($NO_x$, CO and NMVOC). The long-term $CH_4$ RF is mainly affected by the reduction in surface $NO_x$ emissions and it changes very little in the case of the reductions of surface CO and NMVOC alone (probably due to the decrease in OH consumption by CO). This is in agreement with responses from other sensitivity tests performed with UCI CTM by Holmes et al.,[3] with the difference that our short-term aviation $O_3$ RF is not as responsive to surface CO emissions as those modelled with the UCI CTM. In addition, after accounting for the long-term negative RFs that were not given in their 2011 paper (long-term $O_3$ and reductions in stratospheric water vapour, SWV), they also observe that the reduction in surface $NO_x$ emissions decreases the RF from aviation $NO_x$, a 42% reduction in the aviation net $NO_x$ RF resulting from a halving of surface $NO_x$ emissions (C. D. Holmes, personal communication, October 19, 2018), which is in reasonable agreement with the sensitivity shown here. In general, to a great extent, the long-term $CH_4$-mediated effects drive the response of aviation net $NO_x$ RF resulting from modified $NO_x$ emissions. Taking into account that these long-term RFs are fully parametrised as well as the fact that the $CH_4/O_3$ ratio is very model specific[4] make the impact of surface $NO_x$ emissions on aircraft net $NO_x$ RF relatively more uncertain than the impact of other $O_3$ precursor emissions. For example, if the new $CH_4$ RF simplified expression that accounts for short-wave forcing is used[23], reduction in surface $NO_x$ emissions not only decreases the aviation net $NO_x$ RF but also changes its sign from positive to negative. Table 2 gives the recalculated aviation RF numbers from Table 1 using a simplified expression for RF of $CH_4$ as presented by Etminan et al.[23] The improved understanding of $CH_4$ RF has a significant impact on aviation estimates as it increases the negative $CH_4$ RF from aviation $NO_x$ emissions by ~20%, which substantially reduces the aircraft net $NO_x$ RFs (Table 2). Moreover, the revision to the $CH_4$ term provides a perspective that as aviation $NO_x$ emissions are reduced

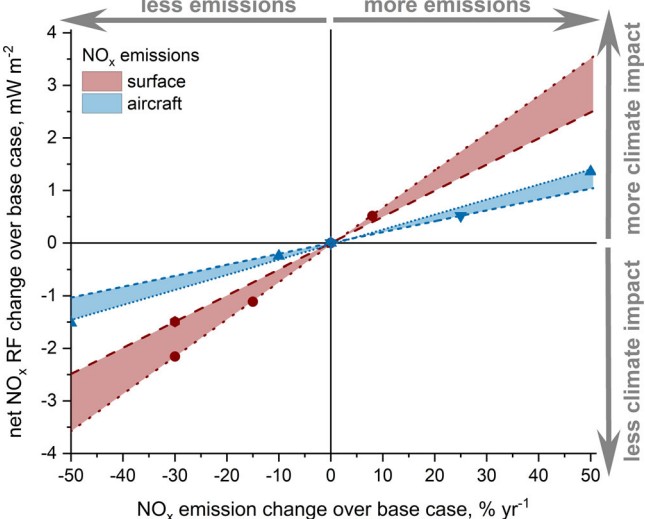

**Fig 3 Aviation net $NO_x$ radiative forcing change versus percentage change in $NO_x$ emission rates from surface and aviation sources.** The aviation $NO_x$ response has been explored for varying, both, surface (red) and aviation (blue) emissions (dots are individual experiments, lines are the best fit lines). In the red case whilst surface $NO_x$ emissions are changing, aircraft $NO_x$ emissions are kept constant and this has been analysed for the highest and lowest projected range of aircraft $NO_x$ emissions, 2050 HighNO$_x$-LowTech scenario (red dashed line; $n = 2$, $r^2 = 1.00$) and 2006 REACT4C (red dotted line; $n = 4$, $r^2 = 1.00$, $p < 0.05$). In the blue case whilst aircraft $NO_x$ emissions are changing, the surface $NO_x$ emissions are kept constant and this has been analysed for the highest and lowest projected range of surface $NO_x$ emissions, 2005 IPCC AR5 (blue dotted line; $n = 4$, $r^2 = 0.998$, $p < 0.05$) and 2050 RCP 2.6 (blue dashed line; $n = 3$, $r^2 = 0.998$, $p < 0.05$). The exact experiments that are used here are presented in Supplementary Table 1.

**Table 2 The recalculated aircraft $NO_x$ radiative forcing (RF) from Table 1 using a revised simplified expression for the RF of $CH_4$ as presented by Etminan et al.[23].**

| Emissions | | | RF, mW m$^{-2}$ | | | | |
|---|---|---|---|---|---|---|---|
| | Background | Aircraft | sO$_3$ | CH$_4$ | lO$_3$ | SWV | Net NO$_x$ |
| 2006 IPCC AR5 | Base | REACT4C | 14.8 | −8.4 | −4.2 | −1.3 | 0.9 |
| | −30% NO$_x$ | | 17.7 | −11.7 | −5.9 | −1.8 | −1.6 |
| | −30% CO | | 14.1 | −8.2 | −4.1 | −1.2 | 0.5 |
| | −30%NMVOC | | 13.9 | −8.3 | −4.1 | −1.2 | 0.3 |
| | −30% ALL | | 16.2 | −10.5 | −5.2 | −1.6 | −1.1 |
| 2050 | RCP 8.5 | Low NO$_x$ High Tech | 45.9 | −28.7 | −14.3 | −4.3 | −1.4 |
| | RCP 4.5 | | 43.7 | −27.8 | −13.9 | −4.2 | −2.3 |
| | RCP 2.6 | | 40.6 | −26.0 | −13.0 | −3.9 | −2.4 |
| | RCP 8.5 | High NO$_x$ Low Tech | 96.3 | −62.2 | −31.1 | −9.3 | −6.3 |
| | RCP 4.5 | | 90.7 | −59.9 | −29.9 | −9.0 | −8.1 |
| | RCP 2.6 | | 83.9 | −56.0 | −28.0 | −8.4 | −8.5 |

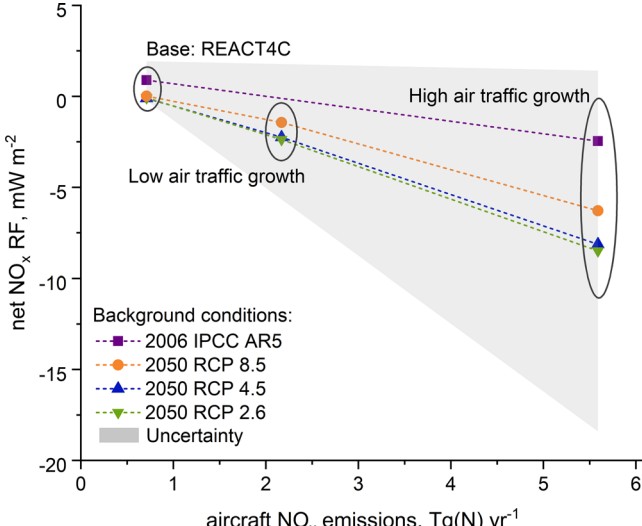

**Fig. 4 Net NO$_x$ radiative forcing by emission rate, updated CH$_4$ parameterisation.** Aviation net NO$_x$ radiative forcing (RF) (the sum of the short-term positive O$_3$ RF perturbation and the negative RF terms caused by a reduction in CH$_4$ lifetime, see Methods), by aviation NO$_x$ emission rate according to a range of background emission scenarios, utilising the updated simplified expression for the calculation of CH$_4$ forcing of Etminan et al.[23] Aviation net NO$_x$ RF systematically decreases with increasing NO$_x$ emissions from aviation, showing a variation according to the background surface emissions, with high mitigation (RCP 2.6) having a smaller aviation net NO$_x$ RF than, lower mitigation scenarios (RCP 4.5, RCP 8.5) for the same aviation NO$_x$ emission. The pattern of behaviour is in contrast to Fig. 1 because the updated CH$_4$ forcing expression accounts for the short-wave forcing of CH$_4$, increasing CH$_4$ RF estimates by approximately 25%, which in the case of aviation net NO$_x$ impacts greatly increases the negative terms from the reduction in CH$_4$ lifetime from aviation NO$_x$, tipping the net NO$_x$ term from being positive to negative with increasing aviation NO$_x$ emissions. Overall uncertainties are indicated by the grey shading, which is one standard deviation (68% confidence interval) from the ensemble of 20 NO$_x$ studies presented in Supplementary Fig. 1.

an *increase in* the global aviation net NO$_x$ RF is shown, and vice versa (Fig. 4 and Table 2). This revised formulation of the CH$_4$ for RF does not contradict findings presented in this study, in terms of the sensitivity of responses, but turns out to be crucial for quantification of net NO$_x$ RFs and it provides a new perspective on the potential RF impact from future aviation NO$_x$ emissions. Other potential effects from NO$_x$ emissions include the direct enhancement of nitrate aerosol and indirect formation of sulfate aerosol (more efficient conversion of sulfur dioxide to sulfuric acid via increased OH). These aerosol effects are associated with large uncertainties and are addressed in only a few modelling studies[24,25] and were not considered here. However, the effects of NO$_x$ on aerosol abundances are expected to result in negative forcings, such that inclusion of these processes would increase the negative NO$_x$-associated forcings and be consistent with the findings of this work that emphasizes the role of the negative forcings.

These new results (Table 2 and Fig. 4) show that as a generality, the net NO$_x$ RF from aviation decreases with air-traffic growth and corresponding increased aviation NO$_x$ emissions with reduced background emissions. The predicted cleaner background in the future acts with these reduced net NO$_x$ RFs. Therefore, it is worth highlighting that the ongoing efforts in cutting ground-level air pollution serve not only air quality improvements but are also beneficial for reducing the climate impact of aviation NO$_x$ emissions.

Climate change is considered to affect tropospheric chemistry. The modified oxidising capacity is expected to influence methane lifetime such that in a wetter and warmer climate it might either shorten or increase[26,27] depending on the scenario. Also, on one hand, the increased water vapour might lead to increased O$_3$ destruction in the tropics, whereas on the other, enhanced stratosphere–troposphere exchange could increase the net O$_3$ flux to the troposphere[14,28]. The impact of the physical aspects of climate change on tropospheric composition is complex and is still highly uncertain. The inclusion of an interactive climate in the experiments might affect the results presented in this study. However, it is not expected that the current findings would become irrelevant, since it is the emission scenarios that to a great extent affect the future evolution of tropospheric chemistry[29,30]. The decreases in surface O$_3$ precursor emissions over present-day values presented in RCP dataset are consistent with most other future emission scenarios that consider even more extensive air quality legislation, e.g. ECLIPSE (Evaluating the CLimate and Air Quality ImPacts of Short-livEd Pollutants) or SSPs (Shared Socioeconomic Pathways) databases. This is in stark contrast to aviation emissions, for which strong growth is predicted and the global civil fleet may more than double within the next 20 years, from ~21,000 aircraft in 2018 to ~48,000 aircraft in 2037[31]. Whilst there is a possibility for the net NO$_x$ RF to be weakened due to a cleaner background, this is not as simple for an aviation CO$_2$ RF. CO$_2$ accumulates in the atmosphere due to its fractional millennial timescale which means that its climate effect is determined by the cumulative emissions over time. As a consequence, the RF in 2018 from aircraft CO$_2$ emissions is around twice greater than that from aircraft NO$_x$ emissions[32]. However, what is even more important is the difference in the nature of their responses (Supplementary Fig. 3). At present, the temperature response from a unit emission of aircraft NO$_x$ is the strongest in the year of emission and it diminishes thereafter. Moreover, after around 15 years it changes the sign from positive (warming) to negative (cooling) to disappear after around 60 years. On the contrary, the emission of CO$_2$ leads to a uniform positive (warming) temperature response from increased atmospheric levels of this gas and after a 100 years, the unit emission emitted in the year 1 still provides a significant positive signal (as modelled by the simple climate model, LinClim, see Methods).

There are various measures to reduce fuel demand (and therefore CO$_2$ emissions) such as market-based measures or stricter aircraft CO$_2$ emission standards; the latter, as it is associated with trade-off between aviation CO$_2$ and NO$_x$ might raise some dilemmas. In view of the low impact of reducing aviation NO$_x$ any potential trade-offs with CO$_2$ should not be risked and also any potential savings in CO$_2$ should not be forsaken in the pursuit of lower NO$_x$ in terms of climate protection[10]. We acknowledge the necessity to reduce aircraft NO$_x$ emissions for local air quality benefits; the source apportionment in any given location is likely to be unique, depending on volume of air traffic and other local sources. However, the aircraft-related emissions of NO$_x$ are of clear importance for many locations. From a climate benefit point of view, we suggest that any vision of more stringent NO$_x$ regulations needs to be revisited, as it might be more worthwhile to concentrate more on CO$_2$ reductions at the cost of NO$_x$, not vice versa, especially in the light of necessary forthcoming decarbonisation to avoid an[33] increase of 1.5°. Coherent comparative assessments that would consider both climate and air quality impacts are needed. There are just a few studies that try to tackle this issue[34–36] and none that would consider these aspects under changing background conditions.

The CO$_2$ emissions still provide the majority of the long-term warming (if not the instantaneous RF) from aviation, and a smaller change in its emission affects the total forcing much more

than an equivalent change in $NO_x$ emission. The mitigation of non-$CO_2$ effects is scientifically uncertain and trading against $CO_2$ could produce perverse outcomes[10], the climate benefits from any reduction of aviation $CO_2$ emissions are indisputable.

## Methods

**Chemistry transport model and emission data.** The model for ozone and related chemical tracers, version 3 (MOZART-3) was used for this study. This is a 3D CTM that has been evaluated by Kinnison et al.[16] and used for an extensive range of different applications[37,38], including studies dealing with the impact of aircraft $NO_x$ emissions on atmospheric composition[39,40].

MOZART-3 accounts for advection based on the flux-form semi-Lagrangian scheme[41], shallow and mid-level convection[42], deep convective routine[43], boundary layer exchanges[44], or wet and dry deposition[45,46]. MOZART-3 reproduces detailed chemical and physical processes from the troposphere through the stratosphere, including gas-phase, photolytic and heterogeneous reactions. The kinetic and photochemical data are based on the NASA/JPL evaluation[47].

The model configuration used in this study includes a horizontal resolution of T42 (~ 2.8° × 2.8°) and 60 hybrid layers, from the surface to 0.1 hPa. The transport of chemical compounds is driven by the meteorological fields from the European Centre for Medium Range Weather Forecast (ECMWF), 6-h reanalysis ERA-Interim data for the year 2006[48]. This meteorological conditions were used for all runs, including the 2050 simulations.

The aviation $NO_x$ emissions for the years 2006 and 2050 were determined based on the REACT4C base case dataset (CAEP/8 movements)[39] and ICAO-CAEP[49] aviation emission projections, respectively. Emissions of aircraft $NO_x$ were calculated to be 0.71 Tg(N) year$^{-1}$ in 2006 and 2.17 Tg(N) year$^{-1}$ in the 2050 low air-traffic growth and optimistic technology-development scenario and 5.59 Tg(N) year$^{-1}$ in the 2050 high air-traffic growth and low technology-development scenario. The 2050 aviation scenarios were chosen to represent the highest and lowest projected range of possible aircraft $NO_x$ emissions in 2050 from data derived from the ICAO-CAEP trends work[49]. First, three aviation traffic demand forecasts out to 2040 are produced (a low, central and high traffic-demand scenario) these demand scenarios are translated by ICAO-CAEP to fleet scenarios, fuel efficiency scenarios are then superimposed upon the fleet scenarios to produce a range of technology and operational improvement scenarios ranging from a technology freeze, through to low, moderate, advanced and optimistic improvement scenarios. Extrapolation of the fuel burn 2040 results out to 2050 is also undertaken and reported by the ICAO-CAEP. Further to the fuel efficiency and traffic demand assumptions, two separate $NO_x$ scenarios were developed by ICAO-CAEP resulting in the derivation of two future 2050 fleetwide $NO_x$ emission indices (EINO$_x$ in terms of grams of $NO_x$ per kilogram of fuel burned): a high and a low EINO$_x$. In this study the high and low EINO$_x$ values for the future fleet are applied to the range of estimates of fuel burn in 2050 to calculate a corresponding range of $NO_x$ emission estimates in 2050. The range of $NO_x$ estimates in 2050 varies from the low $NO_x$ scenario of 2.17 Tg(N) year$^{-1}$ (low EINO$_x$, the low traffic demand with the more efficient optimistic fuel burn scenario, i.e. low fuel burn estimate) and the high $NO_x$ estimate of 5.59 Tg(N) year$^{-1}$ (high EINO$_x$, the high traffic demand forecast and the low fuel efficiency scenario, i.e. higher fuel estimate).

The present-day surface (non-aviation) emissions (base) represented year 2005. The anthropogenic and biomass burning emissions were taken from IPCC AR5[50] and the biogenic emissions were taken from POET[51]. Four different cases were investigated: global reduction of surface $NO_x$ emissions (−30% $NO_x$), global reduction of surface CO emissions (−30% CO), global reduction of NMVOC emissions (−30% NMVOC) and global reduction of ALL these species simultaneously (−30% $NO_x$, CO, NMVOC). All other sources of emissions, including aircraft $NO_x$ emissions were held constant for each experimental case. The 2050 gridded surface emissions (anthropogenic and biomass burning) were determined by Integrated Assessment Models (IAMs) for the three Representative Concentration Pathways[52] (RCPs): a high mitigation scenario that forecast the smallest impact to climate (RCP 2.6), business-as-usual scenario (RCP 4.5) and high climate impact scenario (RCP 8.5). Concentrations of long-lived chemical species and greenhouse gases were based on the RCP emissions, converted to concentrations by Meinshausen et al.[53] Natural emissions (e.g. isoprene, lightning and soil $NO_x$ or oceanic emissions of CO) were not specified in these future scenarios; thus, they were not modified here and were kept the same for all the simulations. The parametrisation of $NO_x$ emissions from lightning was defined as a function of the location of convective cloud top heights[54,55] and their global source were calculated to be 4.7 Tg(N) year$^{-1}$.

The series of sensitivity experiments were performed in order to have a broad perspective on how aircraft and background emissions might affect net $NO_x$ climate impact. The detailed list of simulations exploited in this study shows Supplementary Table 1.

**Radiative forcing calculations.** The Edwards-Slingo radiative transfer model[56] (RTM) was used for the calculation of the forcing associated with aviation $NO_x$-induced short-term $O_3$. The monthly $O_3$ fields from MOZART-3 were converted into mass mixing ratios and interpolated onto RTM vertical and horizontal resolution. The applied RTM is an offline version of the UK Met Office Unified Model

and it calculates the radiative fluxes and heating rates based on the δ-Eddington of the two-stream equations in both, the long-wave (9 bands) and short-wave (6 bands) spectral regions. Cloud treatment is based on averaged International Satellite Cloud Climatology Project (ISCCP) D2 data.[57] Climatological fields of temperature and specific humidity are based on ERA-Interim data[48]. In terms of the 2050 RF calculations, the concentrations of long-lived species were modified according to specific RCP scenarios[53] and were consistent with MOZART-3 set up.

The $CH_4$ concentrations change is assumed to be in equilibrium with the OH change due to the aircraft $NO_x$ perturbation from constant emissions[58]. Since $CH_4$ mixing ratios were prescribed as a lower boundary condition and the simulations were not long enough, the steady-state $CH_4$ concentration ($[CH_4]_{ss}$) was calculated from the change in its lifetime with respect to reaction with tropospheric OH derived from MOZART-3 simulations as shown in Eq. 1:

$$[CH_4]_{ss} = [CH_4]_{ref} \times (1 + 1.4 \times \Delta\tau_0/\tau_{ref}) \qquad (1)$$

A factor of 1.4 was used to reflect the $CH_4$ feedback on its own lifetime[3,58]. Here $\tau_{ref}$ denotes the lifetime of $CH_4$ versus reaction with OH in the unperturbed simulation, $\Delta\tau_0$ the change in $CH_4$ lifetime between the unperturbed simulation and the $NO_x$-perturbed simulation and $[CH_4]_{ref}$ the $CH_4$ mixing ratio in the unperturbed simulation.

This steady-state $CH_4$ aircraft response was further used for the $CH_4$ RF calculations using, as in IPCC AR5[59], a simplified expression (Eqs. 2 and 3) originally defined in Myhre et al.[60]

$$\Delta F = 0.036(\sqrt{M} - \sqrt{M_0}) - (f(M, N_0) - f(M_0, N_0)) \qquad (2)$$

$$f(M, N) = 0.47\ln[1 + 2.01 \times 10^{-5}(MN)^{0.75} + 5.31 \times 10^{-15}M(MN)^{1.52}] \qquad (3)$$

where $N$ is $N_2O$ in ppbv, $M$ is $CH_4$ in ppbv and subscript 0 denotes unperturbed concentrations.

Long-term $O_3$ caused by $CH_4$ changes was calculated according to IPCC AR5[59], as 50% of the $CH_4$ RF. Modified $CH_4$ also affects SWV and give an additional RF of 15% of $CH_4$ RF[61].

The recalculated aviation RF estimates presented in Table 2 are based on the same methodology as original values shown in Table 1 and as described above. The only differences comes from the application of the new simplified expression for RF of $CH_4$[23] as shown in Eq. 4 that replaces the old expression presented in Eqs. 2 and 3:

$$\Delta F = [a \times \overline{M} + b \times \overline{N} + 0.043] \times (\sqrt{M} - \sqrt{M_0}) \qquad (4)$$

where $a = -1.3 \times 10^{-6}$ Wm$^{-2}$ ppb$^{-1}$, $b = -8.2 \times 10^{-6}$ Wm$^{-2}$ ppb$^{-1}$, $M$ and $N$ are concentrations of $CH_4$ and $N_2O$, respectively, and subscript 0 denotes unperturbed concentrations. For terms within the square brackets, the gas concentrations are the mean of the unperturbed and perturbed concentrations, e.g. $\overline{M} = 0.5 \times (M + M_0)$.

**The temperature responses from a unit aircraft $CO_2$ vs $NO_x$ emissions.** The time scales of the climate effects of $CO_2$ and $NO_x$ are very different and these processes of the long-term accumulation vs short-term disappearing, respectively, were explored here. The responses presented on Supplementary Fig. 2 are based on a pulse emission of a 1 Tg(N) year$^{-1}$ in year 1.

In order to observe the temperature response from a unit emissions of aircraft $NO_x$ the methodology presented by Aamas et al.[62] have been applied and Absolute Global Temperature change Potentials (AGTP) have been calculated based on steady-state CTM/RTM results (base case). The forcing for $NO_x$ is assumed to be a result of a pulse that lasts for 1 year followed by an exponential decay of the resulting forcing from the end of the year 1 onwards. The $NO_x$ effect is the sum of the short-term $O_3$ effect, $CH_4$ (with SWV) effect and $CH_4$-induced $O_3$ effect, and there is a perturbation from the forcing for $t < 1$ (this determines the temperature response of the emissions that occur in the first year) and from the forcing for $t \geq 1$ (this determines the temperature response of atmospheric perturbation lasting past one year). Thus, the total AGTP$_{NOx}$ (provided that the time horizon (H) is >1) is the sum of AGTP$_{NOx}^{t<1}$(H) and AGTP$_{NOx}^{t \geq 1}$(H):

a. For perturbation from RF occurring $t < 1$

$$\begin{aligned} AGTP_{NOx}^{t<1}(H) = \Delta F_{NOx}^{SS} \sum_{j=1}^{2} \Bigg\{ & c_j \left[ \exp\left(\frac{1-H}{d_j}\right) - \exp\left(-\frac{H}{d_j}\right) \right] \\ & + \frac{c_j \tau}{\tau - d_j} \left[ \exp\left(-\frac{H}{d_j}\right) - \exp\left(\frac{1-H}{d_j}\right) \exp\left(-\frac{1}{\tau}\right) \right] \Bigg\} \end{aligned} \qquad (5)$$

b. For perturbation from RF occurring $t \geq 1$

$$AGTP_{NOx}^{t \geq 1}(H) = \Delta F_{NOx}^{SS} \left[ 1 - \exp\left(-\frac{1}{\tau}\right) \right] \sum_{j=1}^{2} \frac{c_j \tau}{\tau - d_j} \left[ \exp\left(\frac{1-H}{\tau}\right) - \exp\left(\frac{1-H}{d_j}\right) \right] \qquad (6)$$

The superscript SS indicates steady-state, $\tau$ is the lifetime and it is the short-lived lifetime ($\tau_s$) for short-term $O_3$ (here it is 0.267) whilst the primary-mode lifetime ($\tau_{pm}$) characterises $CH_4$ and $CH_4$-induced $O_3$ (for the MOZART-3's base case it is 12.02 year). The $c_j$ are the components of climate sensitivity and $d_j$ are the corresponding time scales and these values are taken from Boucher and Reddy[63].

In order to observe the temperature response from a unit emission of aircraft $CO_2$ a simple climate model (SCM), LinClim was utilised. LinClim is a linear climate response model that has been customised specifically to aviation[10,64,65]. It uses a single impulse response function that is calibrated against more sophisticated parent model. Aviation fuel data are used to calculate $CO_2$ emissions that is then applied in the linear response function from Hasselmann et al.[66] in order to derive $CO_2$ concentrations. The carbon cycle in LinClim is based on the Maier-Reimer and Hasselmann[67] model and the $CO_2$ RF is calculated using the function applied in IPCC AR4[68]. The temperature response formulation is based on the method presented by Hasselmann et al.[69] The calculated temperature response is also dependent on the climate sensitivity parameter and the lifetime of the temperature perturbation that are tuned to LinClim's parent General Circulation Model (GCM), here it is ECHAM4.

Here the LinClim was used to calculate the temperature response from a pulse/unit emission of aviation $CO_2$ over the long time period. The background scenario chosen was represented by RCP 8.5[53] as its global emissions are the closest to the current levels of $CO_2$ in the atmosphere. The amount of aircraft $NO_x$ that is produced from the fuel burned is described by the emission index (EI$NO_x$). The current EI$NO_x$ is 15.14 g($NO_2$)/kg(fuel)[49] and it has been applied here. Knowing that for every 1 kg of fuel used, 3.16 kg of $CO_2$ is emitted, the 1 Tg of emitted N is equivalent to 217 Tg fuel and therefore, 686 Tg of emitted $CO_2$. This emission was released as a pulse in the year 1 and the consequent $CO_2$ temperature response was observed for the following 100 years.

**Reporting summary**. Further information on research design is available in the Nature Research Reporting Summary linked to this article.

## Data availability
All data discussed in the manuscript and Supplementary Information are presented in Source Data. All data generated for this study (2006 and 2050 CTM and RTM simulations) are available on request from the corresponding author. Source data are provided with this paper.

## Code availability
The modelling data have been post-processed using IDL 8.5.1, and all the scripts are available on request from the corresponding author.

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

## Acknowledgements

This work was supported by the United Kingdom Department for Transport and the European Union's Horizon 2020 Research and Innovation Action ACACIA under grant agreement No 875036.

## Author contributions

D.S.L. conceived of the research, A.S. led the study, R.R.L., L.L.L. and B.O. provided input data, A.S. performed the model simulations and generated all the figures and tables, A.S. and D.S.L. interpreted the results and drafted the manuscript with support from all authors.

## Competing interests

The authors declare no competing interests.
