## [Peer Review File · Nature Communications]

Reviewers' comments:

Reviewer #1 (Remarks to the Author):

This paper applies a chemical transport model to quantify the impacts of changing background precursor emissions on aviation NO_x radiative forcing in the present day (2006) and 2050. The policy interest centres on NO_x stringency and possible trade-offs between NO_x/CO₂ emission changes when increasing fuel efficiency of the aviation sector. NO_x emission increases enhance ozone (climate warming) and decrease methane (climate cooling). The net impact (warming or cooling) can depend on the atmospheric horizontal and vertical location of the NO_x emission. Ozone produced in the UT/LS region is a major climate concern because ozone has a very high longwave radiative forcing efficiency there due to the thermal contrast with the surface. While it is excellent to tackle aviation NO_x emission impacts, I am not convinced that the paper represents a major scientific advance. It is a single model study assessing the sensitivity to emissions changes that may be better placed in an appropriate disciplinary journal.

I have several comments that need to be addressed before publication.

1. The science and policy value of the results depends upon the ability of the model to simulate background NO_x and composition in the UT/LS region and wider atmosphere. The manuscript needs to show validation/evaluation evidence of the skill of MOZART3 in reproducing the absolute concentrations and sensitivity to changes of NO_x in the UT/LS region. The manuscript needs to show comparisons of MOZART3 output with appropriate satellite and aircraft data. For example, there are well-known glaring and persistent biases between observed and modelled NO_x and other trace species in the UT/LS.
2. The net radiative effects of aviation NO_x are tiny (Table 1; 0.002-0.012 mWm²). Are the results based on one year of simulation output from MOZART3? The study needs to calculate the statistical significance of the results relative to interannual climate/meteorological variability. That involves at least ~10 years of simulation output. It is unlikely that the given values in Table 1 are statistically significant relative to interannual climate variability. In some sense, the percentage changes presented i.e. 20-40% may be mis-leading because they refer to a tiny net RF.
3. The paper doesn't mention or discuss any of the recent findings from the USA FAA Aviation Climate Change Research Initiative (ACCRI) e.g. Brasseur et al., 2016: Impact of Aviation on Climate: FAA's Aviation Climate Change Research Initiative (ACCRI) Phase II. *Bull. Amer. Meteor. Soc.*, 97, 561–583, <https://doi.org/10.1175/BAMS-D-13-00089.1>. For example, the reference section of that paper cites several studies with important multi-model updates e.g. Olsen et al., Comparison of model estimates of the effects of aviation emissions on atmospheric ozone and methane, *Geophys. Res. Lett.*, 40, 6004–6009, doi: 10.1002/2013GL057660, 2013. This one: Unger et al., Mid-21st century chemical forcing of climate by the civil aviation sector, *Geophys. Res. Lett.*, 40, 1-5, doi:10.1002/grl.50161, 2013, includes assessment of RCP4.5 and RCP8.5 at year 2050 including emissions changes and climate changes.
4. How does future physical climate change at 2050 influence the aviation NO_x net RF? What happens in the warmer and wetter future world at 2050 for the different RCPs? What is the sensitivity of the aviation forcing to future climate change? The effects of physical climate change on the aviation NO_x net RF may be larger than those due to emissions changes only. It seems weak to present 2050 results for emissions changes only. Those results may become irrelevant or drastically modified under physical climate change.
5. An issue emerging from ACCRI is that NO_x-Ozone-Methane impacts depend on whether a CTM or CCM was used. Fully coupled chemistry-climate models tend to give lower net NO_x RF than the CTMs. Seems to be dependent on stratosphere location relative to aviation NO_x emissions input, and extent of stratospheric chemistry in the model. Is it possible to apply a fully coupled CCM e.g. UKESM1 to calculate the aviation NO_x results? That would represent a scientific advance. We've heard many times over the past 2 decades from studies with one year of CTM output.
6. The results will be sensitive to lightning NO_x. Has the lightning NO_x simulation in MOZART3 been validated against observations and other models e.g. <https://agupubs.onlinelibrary.wiley.com/doi/full/10.1029/2012JD017934> ?

Reviewer #2 (Remarks to the Author):

This manuscript provides insight into the efforts to reduce the climate impact of aviation. The work is careful and seemingly robust. It exposes a new possibility: aviation NO_x reductions depend on the future world and may NOT even be beneficial to the climate. The ideas are new and of very broad interest to the science and policy communities.

The manuscript does need some cleanup before publishing. The discussion of atmospheric chemistry is confusing and not necessarily supportive of the premise.

L33-35: I have trouble reading this sentence. I know what you want to say, but cannot extract it from this.

L48: 'untouched' is odd but maybe. 'unresearched'. or assessed.

L60: 'emission size' makes no sense. Of course one expects the impact to scale somewhat with the aviation source that is not news. If you mean surface emissions, then that is included in the background atmosphere.

L77: need comma after i.e.

L82ff: This phrase is not understandable (L83), and the reaction as stated makes no sense, do you mean HO₂+HO₂?,

L111: 'efficient' is odd here, since the magnitude of NO_x reductions at the surface is HUGE and much greater in TgN/y than is the aviation reduction. The word is wrong, try again.

L123-125: This sentence is very confusing as the effects are not stated in a parallel way: "triggers mostly short-term RF... and long-term RF changes very little" Try for parallel structure, this does not read.

L133ff: Try shorter sentences with fewer clauses.

L149: I cannot understand the "pushes ... harder" at all.

L152ff: At last, here is clearly stated the prime result here. Some of the earlier details need cleaning.

L187: By 'aircraft' I presume you mean aircraft engines, did you mean to include all the operations, including diesel trucks and pushers?

L193: please just drop the "whilst"

Reviewer #3 (Remarks to the Author):

While I don't find any of the results surprising, they nonetheless provide important findings for environmental policy related to aviation. The analysis of the relative merits of decreasing surface emissions of NO_x relative to aviation NO_x is new and has important policy implications for ICAO. For this reason I think it is worthwhile to publish this paper. I have no further concerns or comments.

Reviewer #1 (Remarks to the Author):

This paper applies a chemical transport model to quantify the impacts of changing background precursor emissions on aviation NO_x radiative forcing in the present day (2006) and 2050. The policy interest centres on NO_x stringency and possible trade-offs between NO_x/CO₂ emission changes when increasing fuel efficiency of the aviation sector. NO_x emission increases enhance ozone (climate warming) and decrease methane (climate cooling). The net impact (warming or cooling) can depend on the atmospheric horizontal and vertical location of the NO_x emission. Ozone produced in the UT/LS region is a major climate concern because ozone has a very high longwave radiative forcing efficiency there due to the thermal contrast with the surface. While it is excellent to tackle aviation NO_x emission impacts, I am not convinced that the paper represents a major scientific advance. It is a single model study assessing the sensitivity to emissions changes that may be better placed in an appropriate disciplinary journal.

I have several comments that need to be addressed before publication.

We would like to thank the reviewer for thoughtful comments that undoubtedly helped us to increase the quality of this work. The series of implementations have been done regarding the comparison of modelled data with observations, discussion related to the recent USA FAA ACCRI findings, additional sensitivity analysis showing the rationale of adopted methodology or the validity of the usage of MOZART-3 as a reliable tool for aircraft NO_x analysis. The details of the changes along with the responses to the specific comments are discussed below. We hope that by addressing all these raised issues and dilemmas the reviewer is reassured of the validity and significance of the findings.

1. The science and policy value of the results depends upon the ability of the model to simulate background NO_x and composition in the UT/LS region and wider atmosphere. The manuscript needs to show validation/evaluation evidence of the skill of MOZART3 in reproducing the absolute concentrations and sensitivity to changes of NO_x in the UT/LS region. The manuscript needs to show comparisons of MOZART3 output with appropriate satellite and aircraft data. For example, there are well-known glaring and persistent biases between observed and modelled NO_x and other trace species in the UT/LS.

The ability of MOZART-3 to represent the atmospheric processes and constituents was extensively evaluated by Kinnison et al. (2007) and its capability in reproducing the atmospheric composition with relatively good accuracy was shown in a number of publications (details are given in both the manuscript and SI). Here, the additional analyses have been performed to present the skills of MOZART-3 in reproducing the atmospheric chemistry. The measurements from various sources, stationary stations, ozonesondes, lidar measurements, aircraft campaigns, satellite data have been

utilized to validate CTM's modelled constituents (e.g., CO, NO₂, O₃, PAN) both at ground level and in UTLS region. In general, the magnitudes and temporal variations of NO₂ and CO are well reproduced by the model and there is a good accuracy in reproducing vertical distribution of O₃ in the troposphere and stratosphere for mid- and high latitudes of both hemispheres. However, some discrepancies exist as well (e.g., over tropical tropopause) and they are discussed in detail in Section SI 2.

The new section (SI 2) has been added to SI to presents findings from this analysis where a detailed discussion can be found.

2. The net radiative effects of aviation NO_x are tiny (Table 1; 0.002-0.012 mWm/2). Are the results based on one year of simulation output from MOZART3? The study needs to calculate the statistical significance of the results relative to interannual climate/meteorological variability. That involves at least ~10 years of simulation output. It is unlikely that the given values in Table 1 are statistically significant relative to interannual climate variability. In some sense, the percentage changes presented i.e. 20-40% may be misleading because they refer to a tiny net RF.

This chemical transport model (CTM) MOZART-3 is designed to simulate atmospheric ozone and its precursors. It reproduces detailed chemical and physical processes from the troposphere through the stratosphere, including gas-phase, photolytic and heterogeneous reactions. This CTM is driven with fixed meteorology from reanalysis datasets updated every 6 h that do not incorporate any interactions and feedbacks between chemistry and climate/meteorology; the performed experiments are off-line simulations. Thus, there is no possibility here to calculate the statistical significance relative to interannual climate variability. These modelling settings allow exploring the insights of the chemical processes in detail that brings a useful insight into mechanisms not seen elsewhere.

However, to account for this rather interesting issue, additional sensitivity simulations have been exploited and MOZART-3 has been driven by different meteorological fields representing the years 2004, 2005, 2006 and 2007. It has been observed that this meteorological variability does not affect the O₃ response from aircraft NO_x emissions in a significant way and the global annual averages stay within 2% of the difference. Please see section SI 3 for more details.

The results are based on two years of simulation, where the first year is the spin-up experiment. It has been all explored and the rationale for this methodology has been shown. The additional analysis has been performed that explains why this methodology is appropriate for capturing the short-term O₃ response from aviation NO_x emissions and why there is no need for performing long-term experiments. Please see section SI 3 for more details.

3. The paper doesn't mention or discuss any of the recent findings from the USA FAA Aviation Climate Change Research Initiative (ACCRI) e.g. Brasseur et al., 2016: Impact of Aviation on Climate: FAA's Aviation Climate Change Research Initiative (ACCRI) Phase II. Bull. Amer. Meteor. Soc., 97, 561–583, <https://doi.org/10.1175/BAMS-D-13-00089.1>. For example, the reference section of that paper cites several studies with important multi-model updates e.g. Olsen et al., Comparison of model estimates of the effects of aviation emissions on atmospheric ozone and methane, Geophys. Res. Lett., 40, 6004-6009, doi: 10.1002/2013GL057660, 2013. This one: Unger et al., Mid-21st century chemical forcing of climate by the civil aviation sector, Geophys. Res. Lett., 40, 1-5, doi:10.1002/grl.50161, 2013, includes assessment of RCP4.5 and RCP8.5 at year 2050 including emissions changes and climate changes.

Indeed, this kind of discussion was very limited in the previous version of the manuscript due to space constraints. The current version is much more flexible in that term and the brief 'literature review' is included in the manuscript now. In general, we find an agreement between numbers presented in this study and those available in the literature, given that different future aviation scenarios have been exploited. Please see lines 80-98 in the Result section of the revised manuscript for more details.

4. How does future physical climate change at 2050 influence the aviation NO_x net RF? What happens in the warmer and wetter future world at 2050 for the different RCPs? What is the sensitivity of the aviation forcing to future climate change? The effects of physical climate change on the aviation NO_x net RF may be larger than those due to emissions changes only. It seems weak to present 2050 results for emissions changes only. Those results may become irrelevant or drastically modified under physical climate change.

It has been shown that the future evolution of chemistry strongly depends on emission scenarios (e.g., Shindell et al., 2006, Kawase et al., 2011, Voulgarakis et al., 2013, Young et al., 2013, Szopa et al., 2013, Lu et al., 2019) and that anthropogenic emissions play a more dominant role in determining future tropospheric ozone than does climate change (Liao et al., 2006). The future global ozone predicted with changes in both climate and emissions is generally 12-38% lower (Johnson et al., 1999, Liao et al., 2006) than this simulated with changes in emissions alone. This difference is mainly due to water vapour and temperature increases, so similar dependencies are probable to be observed for methane. Thus, presenting 2050 results only for emission changes is reasonable and scientifically acceptable. Moreover, it makes it comparable with other aircraft studies that also consider only emission changes in the future projections (e.g., Olsen et al., 2011, Unger et al. 2013, Khodayari et al., 2014).

However, indeed the feedbacks exist and the impact of the physical aspects of climate change on the tropospheric composition is very complex. Generally, it is expected that in the wetter and warmer climate, e.g., the increase of water vapour might lead to the ozone reduction (e.g., Brasseur et al., 2006, Young et al., 2013) but the influx of stratospheric ozone might be

promoted (Kawase et al., 2011, Lu et al., 2019), the methane lifetime might shorten (e.g., Shindell et al., 2006, Zheng et al., 2010), as well as it might increase (e.g., Prather et al., 2001, Voulgarakis et al., 2013). Also lightning NO_x emissions are thought to increase that might contribute to ozone production (e.g., Schumann and Huntrieser, 2007, Zheng et al., 2008). All these processes remain still highly uncertain as they depend on the evolution of a multitude climate-related factors.

The inclusion of an interactive climate in the experiments might affect the results presented in this study. However, because the anthropogenic emissions are expected to play a dominant role in determining future levels of tropospheric constituents, it is not expected that the current findings would become irrelevant. This discussion is included in the revised manuscript (see lines 181-190).

5. An issue emerging from ACCRI is that NO_x-Ozone-Methane impacts depend on whether a CTM or CCM was used. Fully coupled chemistry-climate models tend to give lower net NO_x RF than the CTMs. Seems to be dependent on stratosphere location relative to aviation NO_x emissions input, and extent of stratospheric chemistry in the model. Is it possible to apply a fully coupled CCM e.g. UKESM1 to calculate the aviation NO_x results? That would represent a scientific advance. We've heard many times over the past 2 decades from studies with one year of CTM output.

The reviewer is thanked for raising an interesting point. In ACCRI, only a limited number of models took part. In order to address this possibility, we have performed additional analyses of literature studies (normalizing for emissions and any missing terms in the net-NO_x RF term), which are summarized in Figure SI 1, where numbers from both CTMs and CCMs experiments are gathered. This gathering of estimates that represents a wide range of applied models reveals that there are no systematic differences between CTMs and CCMs. Also, it shows the capability of MOZART-3 as a tool for aircraft NO_x study and its advantage by not being an outlier in modelled estimates. An additional discussion on CTM vs CCM has been included in the manuscript, please see the Result section (lines 92-98). The details of this analysis are presented in section SI 3.

We do acknowledge the need for fully coupled-chemistry simulations, as currently, this constitutes a minority (Figure SI 1). However, it is not possible to perform this kind of experiments for this study. Such activity is planned in the near future though, for the forthcoming EU project, Advancing the Science for Aviation and Climate (ACACIA).

6. The results will be sensitive to lightning NO_x. Has the lightning NO_x simulation in MOZART3 been validated against observations and other models
e.g. <https://agupubs.onlinelibrary.wiley.com/doi/full/10.1029/2012JD017934> ?

Indeed, the aircraft NO_x estimates are sensitive to NO_x background (e.g., Holmes et al 2011, Khodayari et al., 2018) and lightning is the major source of NO_x in the upper troposphere. The additional analyses have been performed and lightning NO_x simulated in MOZART-3 has been validated against LIS/OTD climatology datasets. In general, an agreement is found in the pattern of lightning NO_x distribution with some discrepancies in the regional size of the emissions. However, the total global emission of NO_x from lightning in MOZART-3, 4.7 Tg (N) yr⁻¹, is in agreement with the best estimates available in the literature, (5±3) Tg (N) yr⁻¹ (Schumman and Huntrieser, 2007). Please see section SI 2 for a detailed discussion.

Reviewer #2 (Remarks to the Author):

This manuscript provides insight into the efforts to reduce the climate impact of aviation. The work is careful and seemingly robust. It exposes a new possibility: aviation NO_x reductions depend on the future world and may NOT even be beneficial to the climate. The ideas are new and of very broad interest to the science and policy communities.

The manuscript does need some cleanup before publishing. The discussion of atmospheric chemistry is confusing and not necessarily supportive of the premise.

We would like to thank the reviewer for an appreciation of our work and constructive comments that to a great extent helped to improve the manuscript. The discussion of chemistry has been modified and tidied up, hopefully giving more clarity now. The details of all the changes are given below, along with the responses to the specific issues raised by the reviewer.

L33-35: I have trouble reading this sentence. I know what you want to say, but cannot extract it from this.

The sentence has been modified. See lines 34-36.

L48: "untouched" is odd but maybe. "unresearched". or assessed.

The "untouched" has been replaced by "out of discussions". See line 49.

L60: 'emission size' makes no sense. Of course one expects the impact to scale somewhat with the aviation source that is not news. If you mean surface emissions, then that is included in the background atmosphere.

Indeed, it has no use here. The “emission size” was deleted from this sentence. See line 63.

L77: need comma after i.e.

Done. See line 100.

L82ff: This phrase is not understandable (L83), and the reaction as stated makes no sense, do you mean HO₂+HO₂?,

Yes, it meant HO₂+HO₂. The sentence was modified. See lines 103-106.

L111: 'efficient' is odd here, since the magnitude of NO_x reductions at the surface is HUGE and much greater in TgN/y than is the aviation reduction. The word is wrong, try again.

Done. The word “efficient” has been replaced by “plays a crucial role”. See line 131.

L123-125: This sentence is very confusing as the effects are not stated in a parallel way: "triggers mostly short-term RF... and long-term RF changes very little" Try for parallel structure, this does not read.

The sentence was restructured. See lines 144-147.

L133ff: Try shorter sentences with fewer clauses.

Done. See lines 154-156.

L149: I cannot understand the "pushes ... harder" at all.

The sentence has been modified. See lines 168-171.

L152ff: At last, here is clearly stated the prime result here. Some of the earlier details need cleaning.

Thank you for pointing this out. Indeed, this rather important detail should be highlighted earlier. The abstract has been modified to account for these crucial results. See lines 16-18.

L187: By 'aircraft' I presume you mean aircraft engines, did you mean to include all the operations, including diesel trucks and pushers?

By “aircraft” we also assume only aircraft emissions. However, here the word that is used is “airport” and indeed, what we meant here is all the emissions that are associated with the airport infrastructure.

L193: please just drop the "whilst"

Done. See line 222.

Reviewer #3 (Remarks to the Author):

While I don't find any of the results surprising, they nonetheless provide important findings for environmental policy related to aviation. The analysis of the relative merits of decreasing surface emissions of NO_x relative to aviation NO_x is new and has important policy implications for ICAO. For this reason I think it is worthwhile to publish this paper. I have no further concerns or comments.

We would like to thank the reviewer for this very positive comment, hoping that the revised version of the manuscript still constitutes a convincing novelty and importance for a reviewer.

References:

- Brasseur, G. P. et al. Impact of Climate Change on the Future Chemical Composition of the Global Troposphere. *J. Clim.* **19**, 3932-3951 (2006).
- Holmes, C. D., Tang, Q. & Prather, M. J. Uncertainties in climate assessment for the case of aviation NO. *Proc. Natl Acad. Sci.* **108**, 10997-11002 (2011).
- Johnson, C. E., Collins, W. J., Stevenson, D. S. & Derwent, R. G. Relative roles of climate and emissions changes on future tropospheric oxidant concentrations. *J. Geophys. Res.* **104**, D15 18631-18645 (1999).
- Kawase, H., Nagashima, T., Sudo, K., and Nozawa, T. Future changes in tropospheric ozone under Representative Concentration Pathways (RCPs). *Geophys. Res. Lett.* **38**, L05801 (2011).
- Kinnison, D. E. et al. Sensitivity of chemical tracers to meteorological parameters in the MOZART-3 chemical transport model. *J. Geophys. Res.* **112**, D20302 (2007).
- Khodayari, A., Olsen, S. C., Wuebbles, D. J. Evaluation of aviation NO_x-induced radiative forcings for 2005 and 2050. *Atmos. Environ.* **91**, 95-103 (2014).
- Khodayari, A., Vitt, F., Phoenix, D. & Wuebbles, D. J. et al. The impact of NO_x emissions from lightning on the production of aviation-induced ozone. *Atmos. Environ.* **187**, 410-416 (2018).
- Liao, H., Chen, W. T. & Seinfeld, J. H. Role of climate change in global predictions of future tropospheric ozone and aerosols. *J. Geophys. Res.* **111**, D12304 (2006).
- Lu, X., Zhang, L., Shen, L. Meteorology and Climate Influences on Tropospheric Ozone: a Review of Natural Sources, Chemistry, and Transport Patterns, *Cur. Pollution Rep.* **5**, 238-260 (2019).
- Olsen, S. C. et al. Comparison of model estimates of the effects of aviation emissions on atmospheric ozone and methane. *Geophys. Res. Lett.* **40**, 6004-6009 (2013).
- Prather, M. et al. Atmospheric chemistry and greenhouse gases, in: *Climate Change 2001: The Scientific Basis. Contribution of Working Group I to the Third Assessment Report of the Intergovernmental Panel on Climate Change*, edited by: Houghton, J. T. et al., Cambridge Univ. Press, Cambridge, UK, 239-287 (2001).
- Shindell, D. T. et al. Simulations of preindustrial, present-day, and 2100 conditions in the NASA GISS composition and climate model G-PUCCINI. *Atmos. Chem. Phys.* **6**, 4427-4459 (2006).
- Schumann, U. & Huntrieser, H. The global lightning-induced nitrogen oxides source. *Atmos. Chem. Phys.*, **7**, 3823-3907 (2007).
- Szopa, S. et al. Aerosol and Ozone changes as forcing for Climate Evolution between 1850 and 2100. *Clim. Dynam.* **40**, 2223-2250 (2013).
- Unger, N., Zhao, Y. & Dang, H. Mid-21st century chemical forcing of climate by the civil aviation sector. *Geophys. Res. Lett.* **40**, 641-645 (2013).
- Voulgarakis, A. et al. Analysis of present day and future OH and methane lifetime in the ACCMIP simulations. *Atmos. Chem. Phys.* **13**, 2563-2587 (2013).
- Young, P. J. et al. Pre-industrial to end 21st century projections of tropospheric ozone from the Atmospheric Chemistry and Climate Model Intercomparison Project (ACCMIP). *Atmos. Chem. Phys.* **13**, 2063-2090 (2013).
- Zheng, G., Pyle, J. A. & Young, J. P. Impact of climate change on tropospheric ozone and its global budgets. *Atmos. Chem. Phys.* **8**, 369-387 (2008).
- Zheng, G., Morgenstern, O., Braesicke, P., Pyle, J. A. Impact of stratospheric ozone recovery on tropospheric ozone and its budget. *Geophys. Res. Lett.* **37**, L09805 (2010).

REVIEWER COMMENTS

Reviewer #2 (Remarks to the Author):

The authors have done a thorough and fair response to the issues raised in review. They either fixed the ms or explained why it is beyond the scope here. It is ready to be published.

Reviewer #4 (Remarks to the Author):

Comments on "Should we reduce aircraft NOx emissions for the sake of climate?"

The paper is a contribution to a still topical challenge for aviation in trying to mitigate the environmental impact of aircraft emissions on the atmosphere. It concentrates on the impact of NOx emissions on the atmosphere which alter the Earth's radiation budget and in fine the global climate. By using 3D chemistry transport model (MOZART3) and emissions inventories (in 2006 and a prospective year in 2050), the authors discuss the benefit from global warming point of view for reducing NOx compared to CO2 emissions. These simulations confirm and support in general some previous studies performed in the last decade, showing that the NOx impact is less important than thought when formulating ICAO/ACARE objectives for the future. Additionally, as mentioned by the authors, there are many known uncertainties namely on the impact of the non-CO2 emissions, that still remains a big challenge. As an example, the NOx and SOx reactions with other compounds creating new particles (SA) have an impact on climate. Future research is clearly needed that could impact the policy on NOx stringency in a medium/long term. This is not clearly tackled in the ms.

CO2 impact still remains very important for centuries and as reminded by the authors provides the major part of the long-term warming. Hence, fuels without fossil carbon emissions are needed. As such, this work will probably support the ICAO technical committee (CAEP) by providing new data that could potentially assist environmental policy for aviation.

Another point concerning the necessity to decrease the NOx emissions as airports air quality is of concern, as stated by the authors. I don't agree that the major source of NOx is coming from VGA (lines 215-216). Most of the studies on airports air quality performed in Europe (see for example the report on Zurich airport emission inventories) or in North America have shown the opposite i.e. the most important part of NOx is emitted from aircraft engines and APU and not from vehicle ground access (about 30 times greater than those coming from VGA!). What are the authors' sources for this statement? Please clarify.

Aviation is particular in the fact that emissions are emitted from both ground (LTO conditions) and altitude (cruise conditions) that have an impact on air quality as well as climate change. From the engine manufacturer point of view, I am not convinced that it is so easy to separate global warming and air quality objectives. Historically, as reminded in the ICAO environmental report 2019, engine certifications are focused on NOx emitted during LTO cycle (low altitude and human health) and consequently, the relatively large investment on NOx emissions reduction from the engine manufacturers when designing new combustor chamber has been undertaken in this context.

In conclusion, I wonder if the recommendations provided by the author are as relevant as they state. The conclusions appear somewhat definitive, based mainly on prospective data.

Reviewer #2 (Remarks to the Author):

The authors have done a thorough and fair response to the issues raised in review. They either fixed the ms or explained why it is beyond the scope here. It is ready to be published.

Thank you.

Reviewer #4 (Remarks to the Author):

In conclusion, I wonder if the recommendations provided by the author are as relevant as they state. The conclusions appear somewhat definitive, based mainly on prospective data.

We would like to thank the reviewer for thoughtful and constructive comments that to a great extent helped us to increase the quality of this work. Indeed, these points raised by the reviewer, like the implications for air quality-health related issues or the limitations of the modelling, are rather interesting issues. These have been highlighted in the manuscript now and some additional discussions/clarifications have been included. Details of all the changes are given below, along with the responses to the specific issues raised by the reviewer.

The paper is a contribution to a still topical challenge for aviation in trying to mitigate the environmental impact of aircraft emissions on the atmosphere. It concentrates on the impact of NO_x emissions on the atmosphere which alter the Earth's radiation budget and in fine the global climate. By using 3D chemistry transport model (MOZART3) and emissions inventories (in 2006 and a prospective year in 2050), the authors discuss the benefit from global warming point of view for reducing NO_x compared to CO₂ emissions. These simulations confirm and support in general some previous studies performed in the last decade, showing that the NO_x impact is less important than thought when formulating ICAO/ACARE objectives for the future.

The paper not only shows that the net NO_x is less important than CO₂, but especially it highlights the strong dependence of future background conditions on the net NO_x climate impact: the lower surface air pollution, the more beneficial for reducing the climate impact of aviation NO_x emissions it turns out to be. The clean background to some extent might mitigate the warming climate effects resulting from the predicted increasing air traffic. This has been discussed in many places in the manuscript both in the Result and Discussion sections.

Additionally, as mentioned by the authors, there are many known uncertainties namely on the impact of the non-CO₂ emissions, that still remains a big challenge. As an example, the NO_x

and SO_x reactions with other compounds creating new particles (SA) have an impact on climate. Future research is clearly needed that could impact the policy on NO_x stringency in a medium/long term. This is not clearly tackled in the ms.

Thank you for pointing this out. Indeed, the climate impact from coupling NO_x to aerosols is still highly uncertain and unfortunately, it is addressed in only a few studies (e.g., Pitari et al., 2015, Unger et al., 2011). We agree that including these processes might affect the modelling results and we made it clear in the manuscript now. Please see lines 33-38.

CO₂ impact still remains very important for centuries and as reminded by the authors provides the major part of the long-term warming. Hence, fuels without fossil carbon emissions are needed. As such, this work will probably support the ICAO technical committee (CAEP) by providing new data that could potentially assist environmental policy for aviation.

Thank you, we agree with this point wholeheartedly and it is indeed our hope that the work provides policy-relevant information in formulating new emissions regulations, as the reviewer recognizes.

Another point concerning the necessity to decrease the NO_x emissions as airports air quality is of concern, as stated by the authors. I don't agree that the major source of NO_x is coming from VGA (lines 215-216). Most of the studies on airports air quality performed in Europe (see for example the report on Zurich airport emission inventories) or in North America have shown the opposite i.e. the most important part of NO_x is emitted from aircraft engines and APU and not from vehicle ground access (about 30 times greater than those coming from VGA!). What are the authors' sources for this statement? Please clarify.

On reconsideration, we agree with the reviewer that such generalizations are not well-founded and have reformulated the text as follows:

“We acknowledge the necessity to reduce aircraft NO_x emissions for local air quality benefits; the source apportionment in any given location is likely to be unique, depending on volume of air traffic and other local sources. However, the aircraft-related emissions of NO_x are of clear importance for many locations.”

Aviation is particular in the fact that emissions are emitted from both ground (LTO conditions) and altitude (cruise conditions) that have an impact on air quality as well as climate change. From the engine manufacturer point of view, I am not convinced that it is so easy to separate global warming and air quality objectives. Historically, as reminded in the ICAO environmental report 2019, engine certifications are focused on NO_x emitted during LTO cycle (low altitude and human health) and consequently, the relatively large investment

on NO_x emissions reduction from the engine manufacturers when designing new combustor chamber has been undertaken in this context.

Thank you for this useful comment. Indeed, the overall environmental assessment would be ideal for any decision maker. However, it is a challenge to join both climate and air quality-related health objectives; the timescales, locations, metrics, species they all differ depending on the perspective (air quality vs climate) and benefits are usually achieved driven by different factors. The literature is rather vague on this topic and only a few studies deal with this issue (e.g., Dorbian et al., 2011, Mahashabde et al., 2011, Grobler et al., 2019) and definitely this needs more scientific attention in the future.

Like the reviewer mentioned, “the engine certifications are focused on NO_x emitted during LTO cycle (low altitude and human health)” and *by-default* it has been assumed to be good for climate too. And this is one of the rationales of this manuscript, to raise the awareness that it does not necessarily go together. Whatever is good for air quality, does not have to be beneficial for climate and vice versa, especially in the light of the existing trade-off between CO₂ and NO_x.

These aspects are made clearer now and some additional discussion is included in the manuscript, see lines 12-14, 221-224, 227-230.

References:

Dorbian C. S., Wolfe, P. J. & Waitz, I. A. Estimating the climate and air quality benefits of aviation fuel and emissions reductions. *Atmos. Environ.* **45**, 2750–9 (2011).

Grobler, C., et al. Marginal climate and air quality costs of aviation emissions. *Environ. Res. Lett.* **14**, 114031 (2019).

Mahashabde, A., et al. Assessing the environmental impacts of aircraft noise and emissions. *Prog. Aerosp. Sci.* **47**, 15–52 (2011).

Pitari, G. D., et al. Impact of coupled NO_x/aerosol aircraft emissions on ozone photochemistry and radiative forcing. *Atmosphere* **6**, 751–782 (2015).

Unger, N., Global climate impact of civil aviation for standard and desulfurized jet fuel. *Geophys. Res. Lett.* **38**, 1–6 (2011).

REVIEWERS' COMMENTS

Reviewer #4 (Remarks to the Author):

The responses to the reviewer comments/questions were globally satisfactory. As a consequence, the reviewer recommend the publication of this modeling study in nature communications.